# Design of a Remote, Multi-Range Conductivity Sensor

**DOI:** 10.3390/s23249711

**Published:** 2023-12-08

**Authors:** Georgiana Dima, Anna Radkovskaya, Christopher J. Stevens, Laszlo Solymar, Ekaterina Shamonina

**Affiliations:** Department of Engineering Science, University of Oxford, Oxford OX1 3PJ, UK

**Keywords:** conductivity detection, remote sensing, low conductivity

## Abstract

So far, research on remote conductivity detection has primarily focused on large conductivities. This paper examines the entire conductivity range, proposing a method that can be adapted to the desired application. The optimization procedure for the different regions is presented and discussed. Specific interest is given to the low-conductivity range, below 10 S/m, which covers human body tissues. This could lead to applications in body imaging, especially for induction tomography. Conductivities below 12.5 S/m are extracted experimentally with an error below 10%.

## 1. Introduction

Eddy current analysis is an established non-contact sensing technique [1,2]. A wide range of antenna and metamaterial arrays [3,4,5,6] has been developed; these arrays are involved in eddy current sensing techniques. Most publications in the field focus on non-destructive testing (NDT) applications, which include thickness measurements [7,8,9,10,11,12,13,14,15,16,17], detection of cracks and defects [18,19,20,21,22,23,24,25,26,27,28], as well as measurements of conductivity σ [10,11,12,13,14,29,30,31,32] and permeability μ [33,34,35] in highly conductive samples. These applications cover only a fraction of the conductivity spectrum, although detecting mid-range and low σ can lead to exciting applications.

For the mid-range conductivity spectrum, a recent area that has elicited interest is in creating conductive 3D-printed materials [36,37,38,39,40,41,42,43,44,45,46]. Once a 3D sample has been printed, it is hard to assess if the material has the desired properties without potentially damaging the sample. For example, the conductivity of Electrifi, one of the most widely used 3D-printed filaments, can be heavily influenced by printing conditions, but this can only be assessed through invasive methods. The conductivity of these 3D-printed materials ranges between 10 and 50,000 S/m, which in this paper will be broadly classified as the mid-range σ space. The conductivities of various 3D-printed filaments used in the literature are described in [41].

Another area with significant potential is that of biomedical imaging. Inductive tomography [47,48] is a new, promising imaging method where transmitting and receiving coils are used to map the conductivities and permittivities of different body tissues and, ultimately, image through them or monitor any abnormalities. One such application is detecting the presence of water in the lungs. At 50 MHz, as shown in Table 1, the conductivity range for the lung is between 0.28 S/m (when the lung is inflated) and 0.52 S/m (when the lung is deflated). If water is present in the lungs, this conductivity rises to around 0.5 S/m (for the inflated lung) and 0.7 S/m (for the deflated lung) [49,50,51]. Such differences can be detected by the sensor and lead to further tests being performed on the patient. The applicability of eddy current testing in a similar area is shown in [52,53], where the breathing cycle and low oxygen concentrations are monitored. Currently, more in-depth research is required to optimize and transform this idea into functional equipment. Although several publications have investigated high-conductivity detection, few have investigated methods of detecting low conductivity. Yin et al. [31] investigated the conductivity and the thickness of a salty water solution and managed to detect these properties with an accuracy of 3%. However, the range of conductivities considered was relatively narrow, with only four considered values (3.7, 6.4, 8.3, 10.2 S/m). Additionally, a scaling factor was applied in [31] to ensure that the analytical model used matched the experimental values. Permittivity was not considered; however, for low σ in the biomedical application discussed here, this parameter is essential.

The present paper focuses on identifying a design procedure for probes that will detect σ in each of the three aforementioned cases (low, mid, and high range σ) and will present results that support the design. Specific focus will be given to detecting conductivities similar to those in the human body, as novel applications can emerge from being able to differentiate tissues depending on their conductivity. The conductivity and permittivity of some biological tissues at the operation frequency of 50 MHz are presented in the table below (Table 1) as obtained from [49,50,51].

First, the analytical model and the working principle of the probe will be presented in Section 2.1. The setup used for the numerical simulations in CST Microwave Studio will be introduced in Section 2.2, while the experimental setup and the data post-processing will be described in Section 2.3. In Section 3.1 and Section 3.2, the analytical results for various system parameters will be provided in order to justify the design criteria suggested. The focus will then shift toward the low conductivity range in Section 3.3, where the conductivities of water and salt will be extracted from experiments. The conclusion of the paper will be outlined in Section 4, where final recommendations for the design procedures will be offered.

## 2. Materials and Methods

### 2.1. Analytical Model

The sensing probe used relies on a single resonating element with one loop of radius $r_{0}$ and wire thickness *w*, tuned to a resonating frequency $f_{0}$ using a lumped capacitor. The probe is placed a distance *h* above a testing material of thickness *t*, conductivity, σ, relative permittivity $\varepsilon_{r}$, and relative permeability $\mu_{r}$. The testing sample dimensions in the horizontal plane are expected to be much larger than the probe diameter. This configuration is shown schematically in Figure 1.

When the resonator is excited, a magnetic flux proportional to the current flowing through the circuit is generated. The constant of proportionality between the magnetic flux and the current is called the self-inductance (*L*) of the element. If the coil is placed near a material with different electromagnetic properties than air, a time-varying magnetic flux will generate eddy currents within the material. This changes both the magnitude and the direction of the overall magnetic flux, resulting in a complex self-inductance. In other words, the presence of a testing material beneath an excited coil changes the coil’s self-inductance. Figure 2 shows the absolute value and orientation of the magnetic field in the plane y=0, as obtained from analytical simulations for different values of σ (0.1, 10, 1000 S/m). Looking at the field strength at z=−50 mm, it is evident that as the conductivity increases from 0.1 S/m to 10 S/m and 1000 S/m, the magnitude of the field strength drops from 1.33 A/m to 0.5 A/m and then to $8.9\times 10^{-10}$ A/m. This supports the fact that, as conductivity increases, the depth of penetration decreases, and so does the magnitude of the field. The overall orientation of the magnetic field is also altered by the presence of a conductive sample. This is the result of the mirroring field from the surface of the sample, which cancels out some of the field within the coil. This will be further explained in Section 3.1, where the similarities between a mirror and an ideal conductor will be presented.

This change can be found by generalizing the partial differential equation used by Dodd and Deeds [54] for vector potential A due to an applied current density $i_{0}$:(1)$$\nabla^{2}A=-\mu i_{0}+\mu \sigma \frac{\partial A}{\partial t}+\mu \varepsilon \frac{\partial^{2}A}{\partial t^{2}}+\varepsilon \partial \left(\frac{1}{\mu }\right)\times \left(\nabla \times A\right)$$

Equation (Equation 1) is derived from Maxwell’s equations [55], ensuring full applicability. The geometry in Figure 1 is divided into four layers (I–IV), which are isotropic, linear, and homogeneous. The excitation is considered filamentary and is situated between layers I and II. Simplifications similar to those in [54] are performed on Equation (Equation 1). The main difference is that, here, as in [56,57], the term $\mu \varepsilon \;\delta^{2}A/\delta t^{2}$ is considered relevant and accounted for in the analytical model. For high conductivity and small frequencies, this term is usually ignored in the literature. Using this term will allow for an accurate estimation of low conductivities. Once the simplifications are made, the equation is solved and the boundary conditions between every two layers are used to determine the vector potential in each region. Here, the self-inductance of a coil is approximated to be equivalent to the mutual inductance between two filaments, spaced a geometric mean distance ($R_{\mathrm{GMD}}$) apart, similar to Maxwell’s approach for calculating self-inductance at higher frequencies [58]. $R_{\mathrm{GMD}}$ can be found in [59,60,61]. The full derivation of the vector potential and the calculation of self-inductance is provided in Appendix A. The final expression for the inductance is as follows:(2)$$L=L_{r}+jL_{i}=\pi r_{0}^{2}\mu_{0}\int_{0}^{\infty }\alpha_{0}J_{1}^{2}\left(\alpha r_{0}\right)\left(e^{\alpha_{0}R_{\mathrm{GMD}}}+\frac{\left(\alpha_{0}^{2}-\alpha_{1}^{2}\right)e^{-\alpha_{0}2h}\left(e^{2\alpha_{1}t}-1\right)}{{\left(\alpha_{1}+\alpha_{0}\right)}^{2}e^{2\alpha_{1}t}-{\left(\alpha_{0}-\alpha_{1}\right)}^{2}}\right)d\alpha$$

Two important parameters of the coils will be simulated, following the inductance calculation: the resonant frequency (denoted as $f_{0}$ in free space and $f_{\mathrm{res}}$ in the vicinity of a material) and the quality factor ($Q_{0}$ in free space and *Q* in the vicinity of a material). In free space, these parameters are given by
(3)$$f_{0}=\frac{1}{2\pi \sqrt{LC}}$$
(4)$$Q_{0}=\frac{2\pi f_{0}L}{R}$$

When a material is present, the derivation becomes somewhat more complicated, but a reasonable approximation is found to be
(5)$$f_{\mathrm{res}}=\frac{1}{2\pi \sqrt{L_{r}C}}$$
(6)$$Q=\frac{2\pi f_{\mathrm{res}}L_{r}}{R-j2\pi f_{\mathrm{res}}L_{i}}$$

The circuit analysis shows that the real part of the new inductance, $L_{r}$, becomes the new inductance of the coil, while the imaginary part, $L_{i}$, contributes to the resistance in proportion to the resonant frequency.

### 2.2. Numerical Model

CST Microwave Studio 2020 was used as a numerical tool to offer more points of comparison for the analytical and experimental results. Figure 3 shows the CST model used for the simulations that were performed here. The mesh distribution is displayed in the figure. The bounding box that is filled with air is 500 mm away from the structure in all directions and the boundary conditions are open (add space). The background mesh is set at 5 cells per wavelength. The local mesh for the coil is set at 0.5 mm, and for the sample, it is 15 mm, The coil is tuned using a lumped capacitor, placed in series with the S-parameter excitation source, which has an impedance of 50 Ohms. The sample is designed to be six times wider than the coil diameter, as this size has been determined to be large enough for the sample to be approximated as infinite, thus comparable with the analytical model, yet small enough to allow for computation within a reasonable timeframe. The accuracy of the S-parameter measurements is set at 1.5%. The $Z_{11}$ parameter is extracted and then the resonant frequency is extracted as the frequency at which $Z_{11}$ reaches a minimum, while the quality factor is obtained by calculating the ratio between the bandwidth of the $Z_{11}$ curve and the resonant frequency.

### 2.3. Experimental Setup

Figure 4a shows the schematic of the experimental arrangement and Figure 4b–e display some photographs of the experiment. The coils used were created by winding a single loop of wire into a coil-shaped mold that was 3D-printed with high-impact polystyrene (HIPS). This material was chosen for its low dielectric constant, ensuring that the structural support would not affect the overall results. To secure the coil in place, a sheet of Rohacell foam was utilized. Given its properties, very similar to those of air, this foam is not expected to significantly impact the results either. The coil, with a radius of $r_{0}=25$ mm and wire width of w=1.04 mm was placed at a distance of h=5.32±0.11 mm from the sample, which is represented by a water tank measuring 300×300 mm, with a water layer thickness of t=100 mm. The conductivity of the water was gradually changed by adding salt. The conductivity after each salt-adding iteration was measured using a Hanna HI-8633 multi-range conductivity meter.

The coil is connected to an 8753ES Hewlett Packard Vector Network Analyzer (VNA) through the SMA connector and a 50Ω cable whose impact was removed through a standard calibration procedure. The $S_{11}$ parameter was obtained for 1601 points in the frequency spectrum. The power used was 0 dBm and the number of averages for each data point was 2. For each σ, 20 data points were obtained. The $S_{11}$ parameter was used to extract the impedance spectrum, $Z_{11}$, through the following formula, where $Z_{L}$ is the known impedance and 50Ω is the load of the VNA:(7)$$Z_{11}=Z_{L}\frac{S_{11}+1}{S_{11}-1}$$

The data extracted are then filtered by using a Savitzky–Golay filter with a window length of 32 and order 2, with 10,000 values interpolated in the frequency range considered in order to increase the accuracy of Q estimation. The resonant frequency $f_{\mathrm{res}}$ is extracted as the frequency where the impedance parameter reaches a minimum and the quality factor *Q* is obtained by dividing this resonant frequency through the 3 dB bandwidth, BW. In this way, the free space properties of the coil were estimated to be $f_{0}=49.777$ MHz and $Q_{0}=256$. Once $f_{\mathrm{res}}$ and *Q* are extracted for different conductivity samples, depending on the value of $f_{\mathrm{res}}$, one of them becomes the main extraction parameter. The extraction is performed using the bisection method, where the conductivity of the analytical model is changed until the analytical value of the extraction parameter matches the experimental one. Two extraction parameters are proposed, namely the quality factor and the resonant frequency for which an analytical model is provided in Section 2.1. Section 2.1 and Section 3.1 discuss the parameters that should be used for extraction in each range, while Section 3.2 describes the proposed extraction procedure.

## 3. Results

### 3.1. Q and $f_{\mathrm{res}}$ Analytical

Figure 5 shows the variation of $f_{\mathrm{res}}$ and *Q* with conductivity. The coil has $r_{0}=25$ mm, w=1 mm, $f_{0}=50$ MHz, $Q_{0}=250$, and the sample used (t=10 cm and $\mu_{r}=\varepsilon_{r}=1$) changes its conductivity in the range of 0.01 S/m to $10^{8}$ S/m. The coil is placed at a distance of h=5 mm. As σ increases, $f_{\mathrm{res}}$ starts increasing steadily and then plateaus to a constant value. This constant value is equivalent to the case of an ideal conductor that behaves like a mirror due to the fact that the depth of penetration, known as the skin depth, is very narrow. The reduction in the depth of penetration is clearly presented in Figure 2, where a conductivity of 1000 S/m results in the magnetic field only being significant at the top of the surface. In the case of an ideal conductor, the magnetic field is fully canceled out as a consequence of Lenz’s Law, and the system can be interpreted as containing two coils placed axially in free space, a distance of 2h=10 mm apart. However, because the conductor acts as a mirror, only the asymmetrical mode for the coupled resonators is supported. In the case of axially coupled resonators, this mode corresponds to the upper peak in the current spectra. The position of this peak is, therefore, the position where the resonance of a coil above a sample with very large conductivity will settle. This correlation is clearly illustrated in Figure 6. The figure displays the normalized absolute value of the current spectra for a single coil placed above a conductive sample with $\sigma =10^{8}$ S/m (red dotted line) and the normalized absolute value of the current spectra for one of the coils in a system of two coils placed axially in free space a distance 2h apart (black line). The mutual inductance for the coil is calculated in a similar manner as the self-inductance by assuming that the coils are filamentary and the distance between them is no longer $R_{GMD}$, as in Section 2.1, but 2h. The upper peak of the coupled system perfectly matches the resonant peak in the single-coil system placed above a nearly perfect conductor. In the case of non-ideal conductors, the reflection coefficient is lower than 1, leading to a combination of transmission and reflection and, hence, intermediary resonant frequency values.

Moving on to the quality factor: Q starts high for very low conductivities and then drops in the mid-range, where it remains relatively steady. This constant region is followed by an increase in Q for high-conductivity values. This behavior is a consequence of two competing phenomena. The first one concerns energy dissipation in the sample material due to the generated eddy and displacement currents. These currents increase as the conductivity increases. The second phenomenon is the decline of the skin depth as the conductivity increases. This means that the regions where currents are created become smaller, leading to lower losses. For low σ, the first phenomenon is dominant, and, as conductivity increases, the second mechanism becomes more relevant, ultimately being the dominant of the two. The conductivity where the transition occurs depends on multiple factors, such as the thickness of the sample, the operating frequency, and the geometrical parameters of the split-ring resonator.

From an extraction perspective, these two parameters cover the entire conductivity range. *Q* changes sharply in the low and high ranges while $f_{\mathrm{res}}$ changes significantly in the mid-range of σ. The decision on where to switch between the two can be made depending on $f_{\mathrm{res}}$. For a specific known geometry, the two frequencies at which the transition between the two detection methods occurs can be calculated analytically, depending on the specific accuracies that can be achieved in extracting $f_{\mathrm{res}}$ and *Q*.

As mentioned above, these curves are impacted significantly by certain parameters. One of these critical parameters is the thickness of the sample. Figure 7 shows the variation in resonant frequency and quality factor as σ varies in the range of 0.01 S/m to $10^{8}$ S/m for four values of sample thicknesses (0.5, 2, 10, 100 mm). The parameters of the system are the same as in Figure 5, except for the thickness that takes the aforementioned values. The dotted lines represent the positions where the skin depth is the same as the thickness of the sample. In free space and for very large conductivities, the curves have the same characteristics. As the thickness becomes smaller, larger conductivities are required in order to notice changes in the resonant frequency. This is because, at low conductivities, the magnitude of the generated eddy currents is lower, and so is the accessible depth of penetration due to the limited sample thickness. This results in lower overall perturbations from the free-space scenario. When the depth of penetration decreases to the skin depth due to the increase in σ, the curves for both resonant frequency and quality factor start converging to the curve for t=100 mm. From the quality factor perspective, a thinner sample requires higher current densities (hence, higher conductivities) for the transition between the two phenomena mentioned above to occur because the depth of penetration is limited to the thickness of the sample. Overall, thinner specimens results in less resonant frequency and quality factor changes for conductivities with skin depth larger than *t*.

Another important parameter is the ratio between the height, h, and the radius of the coil, $r_{0}$. It is important to note that, even when decreasing the height and increasing the coil radius lead to similar changes in the studied parameters, experimentally, increasing the radius will lead to an increase in $Q_{0}$, which is not accounted for here. Figure 8 shows the variation of resonant frequency and quality factor with σ as the ratio takes the values 0.2, 0.5, 1, and 2. All other parameters are the same as in Figure 5. As expected, moving the sample closer results in larger changes in both monitored parameters. In the case of resonant frequency, for very high σ, this can be seen as an increase in the coupling coefficient between the real coil and the imaginary mirrored coil. For the quality factor, the transition region shifts to lower σ the further away from the sample we move. This is because the depth of penetration is not significantly influenced by the proximity, but the current densities are. If the coil is placed further away, the transition will occur at a larger depth of penetration (hence, lower σ), as the current densities to be counterbalanced are not as large. This figure is significant from an extraction perspective as it shows that proximity is favorable for large changes which can then be easier to monitor. However, it is important to note that moving closer to a sample is experimentally challenging as stray unpredictable capacitances can be generated.

The last parameter to be discussed is the free-space resonant frequency of the split-ring resonator. Figure 9 shows the variation of the normalized resonant frequency and quality factor with σ for 10, 20, 50, and 100 MHz. All other parameters are the same as in Figure 5. From the resonant frequency perspective, increasing the resonant frequency shifts the curves to the left, meaning that changes occur for lower conductivities. Going back to the two-coil system analogy, this is intuitive, as the sample is easier to approximate as an ideal conductor when the frequency is larger because the skin depth decreases with frequency. The same shift to the left is applicable to the quality factor plot, with the same underlying explanation: the depth of penetration decreases more quickly, hence the transitions between the two competing phenomena occur at lower conductivities. From an extraction perspective, changing the resonant frequency can be one way to target different σ ranges, as it offers a method of transposing the plots without interfering with the magnitude of the changes.

### 3.2. Simulated Extraction Results

Figure 10 shows the extraction error for the entire σ range when an artificial error of 0.01% is present in estimating $f_{\mathrm{res}}$ and 0.5% in estimating *Q*. These values are in line with the variance noticed when the parameters are measured experimentally, multiple times for the same sample. The parameters of the system are $r_{0}=25$ mm, w=1 mm, h=5 mm, t=10 cm, $f_{0}=50$ MHz, $Q_{0}=250$. The extraction was performed using $f_{\mathrm{res}}$ and *Q*, which are represented using the transparent black and red lines, respectively. The minimum error between the two extractions is then chosen. There are two breakpoints that represent the transition between the two types of extraction. In the real system, these two breakpoint frequencies will be stored for the specific geometry. If the experimental frequency is below the first breakpoint or above the second breakpoint, Q will be used for extraction; otherwise, $f_{\mathrm{res}}$ extraction will be performed.

Figure 11 shows the extraction for different values of free-space resonant frequency, $f_{0}$ (Figure 11a), and for different values of the distance to the radius ratio (Figure 11b). Starting with Figure 11a, it is noticeable that lower free-space resonant frequencies, $f_{0}$, result in better detection of large conductivities. The blue curve, corresponding to $f_{0}=10$ MHz, is capable of detecting σ up to $10^{8}$ S/m with an error lower than 10%. However, when looking at the low σ range, $f_{0}=10$ MHz results in significant errors for conductivities below 0.1 S/m, while higher frequencies (namely 20 MHz—orange and 50 MHz—green) can detect (with an accuracy above 95%) conductivities down to 0.1 S/m. This is in line with the conclusions from Section 3.1. The same is true for Figure 11b, which shows the detection error for different height-to-radius ratios. As the probe is placed closer to the conductive material, the error decreases, and the detection range increases, especially toward the higher σ range. This is supported by Section 3.1, where it was shown that decreasing the distance between the probe and the sample results in larger changes, making them easier to monitor. In conclusion, Figure 11 displays two important characteristics of the probe under study. The first one is the shift in the σ detection range toward high conductivities as the frequency is decreased, and the second one is the improved detection that results from minimizing the distance between the sample and the coil.

### 3.3. Experimental Results for Low Conductivity Samples

#### 3.3.1. Comparison of f and Q

Figure 12 shows the variation of $f_{\mathrm{res}}$ and *Q* as conductivity is increased up to 20 S/m in the system described in Section 2.3. The data are presented for the analytical model (blue solid line), the raw experimental data (red data points), the smoothed experimental data (blue data points), and CST Microwave Studio data (black dashed line). The experimental data points are presented using error bars for which the mean and the variance were obtained from the 20 experimental points taken for each σ. From the figure, it can be seen that the variance is significantly decreased once the data are smoothed. It is important to note that the data were normalized to the first point corresponding to water without salt (σ=0.04 S/m). This normalization is necessary because the water layer interacts with the metallic plate of the SMA connector, which increases the capacitance of the coil. The same normalization was performed for the CST data to ensure that the first point in the σ sweep corresponds to the first point in the experimental data. It can be seen that the match is very good, indicating that conductivity detection is possible.

#### 3.3.2. Extraction Results

Figure 13 shows the results of extracting conductivity from the experimental data using the analytical model (red line) and the CST data (blue dots). The red transparent filled-in curve represents the entire range of conductivities extracted when one takes into account the 0.11 mm height uncertainty. The extraction from CST data was conducted by interpolating between the pre-stored CST values. All the extractions were performed using *Q* as the main extraction parameter. The analytical model performs better in extraction compared to the CST one. Conductivities of up to 12.5 S/m are detected with an error lower than 10% while conductivities of up to 5.5 S/m are detected with an error of below 5%. Most importantly, using the quality factor as a separation criterion proves to be an effective technique for distinguishing between very low σ samples. As shown in Table 2, the mean error in detection for conductivities lower than 1 S/m is 4.5%.

Table 2 compares the mean performance of the analytical extraction with the interpolation extraction using CST for different ranges of conductivity. For each range of conductivity, the number of samples considered is shown in the second column. While the analytical model has a mean error lower than 5.5% for the entire range considered, the error using the numerical solver is higher (almost 10%).

The extraction error is also influenced by the accuracy with which we can estimate the height, *h*. When the height, *h*, is deviated by 0.11 mm (approximated to be the error in estimating the distance for the presented experiment), the mean extraction error for the analytical model goes from 5.5% to 6.6%. This suggests that accurate height measurements should be prioritized when implementing the method.

## 4. Discussion and Conclusions

A method for remotely detecting conductivity using the quality factor and the resonant frequency of a single split-ring resonator is presented. The analytical model used for this method is outlined and the choice of coil parameters is discussed using analytical parametric plots. In order to optimize detection, it is suggested that the coil radius be as large as possible and the distance from the sample is as small as possible. The free space resonant frequency of the coil should be low (in the range of 1–10 MHz) for high conductivity samples and larger (in the range of 50–100 MHz) for medium and low conductivity ones. Experimental data are presented for a sample with low conductivity and detection with an error below 10% is achieved for conductivities below 12.5 S/m. It is concluded that detection in the full range is possible and that detecting conductivities down to 0.1 S/m is achievable. More importantly, it is proven that using this method and the chosen optimization parameters, one can distinguish between samples with very low conductivities, such as human tissue. Separating different types of body tissue conductivities is highly relevant as it enables research into abnormalities detection (such as water in the lungs) and holds significant importance for body imaging, particularly in induction tomography. Considering that very little power is used for sensing (1 mW), the specific absorption rates inside human tissues are lower than 0.1 W/kg. This is significantly below the safety limit of 2 W/kg, making further development of such a method for body imaging attractive from a safety perspective.

## Figures and Tables

**Figure 1 sensors-23-09711-f001:**
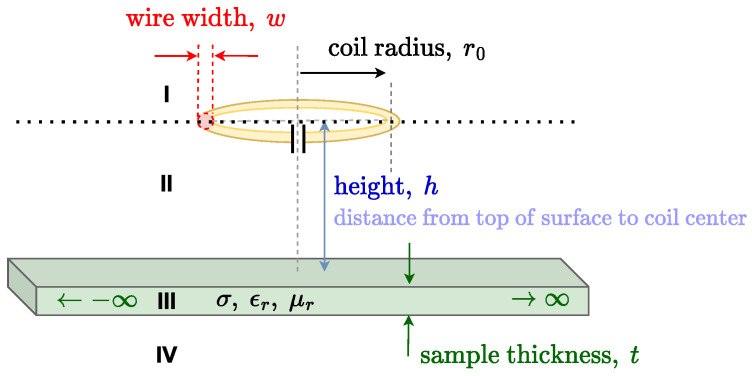
Schematic diagram of the system showing all the relevant geometrical parameters. Regions I–IV represent the regions where the material properties change. Between these regions, boundary conditions are imposed as described in Section 2.1.

**Figure 2 sensors-23-09711-f002:**
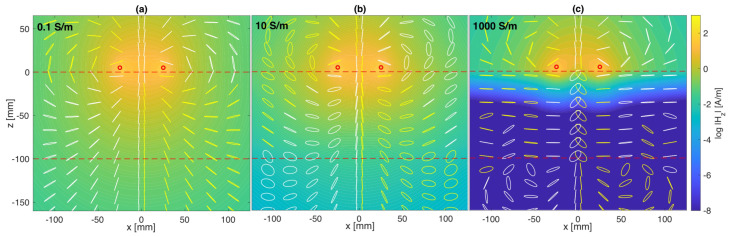
Logarithmic value of magnitude and polarization of the magnetic field, *H*, for a system with a coil of radius, $r_{0}=25$ mm, coil width, w=1 mm, resonant frequency, $f_{0}=50$ MHz, free-space quality factor, $Q_{0}=250$, height, h=5 mm, sample thickness, t=10 cm, and relative permittivity and permeability, $\mu_{r}=\varepsilon_{r}=1$. The conductivity of the sample is: (**a**) 0.1 S/m; (**b**) 10 S/m; (**c**) 1000 S/m.

**Figure 3 sensors-23-09711-f003:**
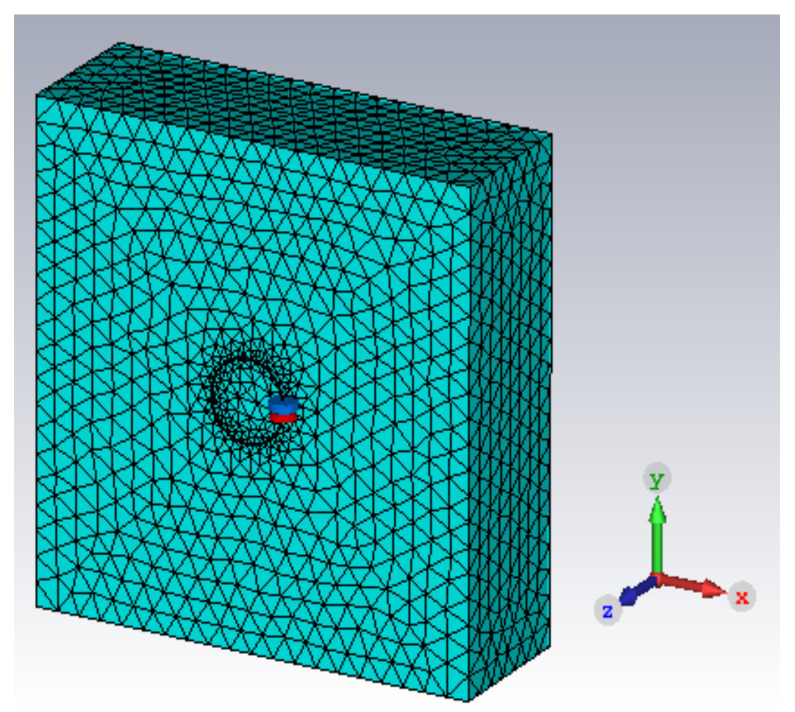
CST Microwave Studio model with the shown mesh. The present components are the conductive sample, the coil, the tuning capacitor, and the excitation port, placed in series with the capacitor.

**Figure 4 sensors-23-09711-f004:**
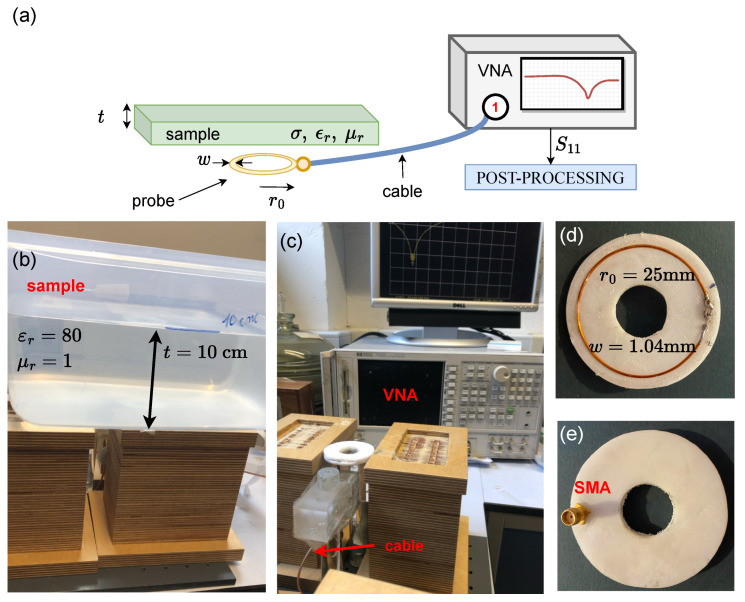
Experimental arrangement: (**a**) schematic representation, (**b**) sample photograph, (**c**) photograph of the coil and VNA with no sample, and (**d**,**e**) top and bottom photographs of the probe.

**Figure 5 sensors-23-09711-f005:**
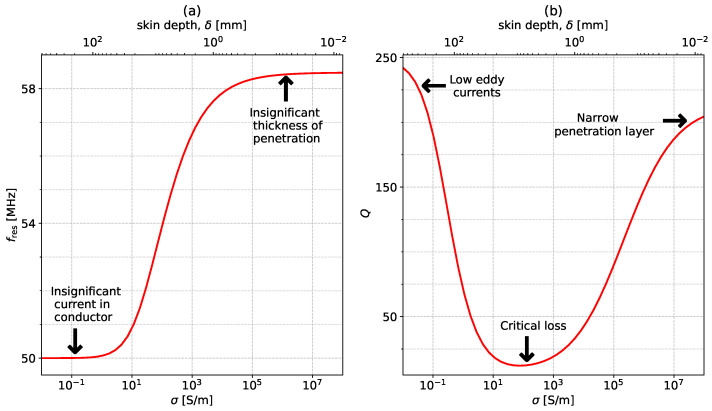
Variation of (**a**) $f_{\mathrm{res}}$ and (**b**) *Q* with conductivity, σ. The system parameters are $r_{0}=25$ mm, w=1 mm, $f_{0}=50$ MHz, h=5 mm, $Q_{0}=250$, t=10 cm, and $\mu_{r}=\varepsilon_{r}=1$.

**Figure 6 sensors-23-09711-f006:**
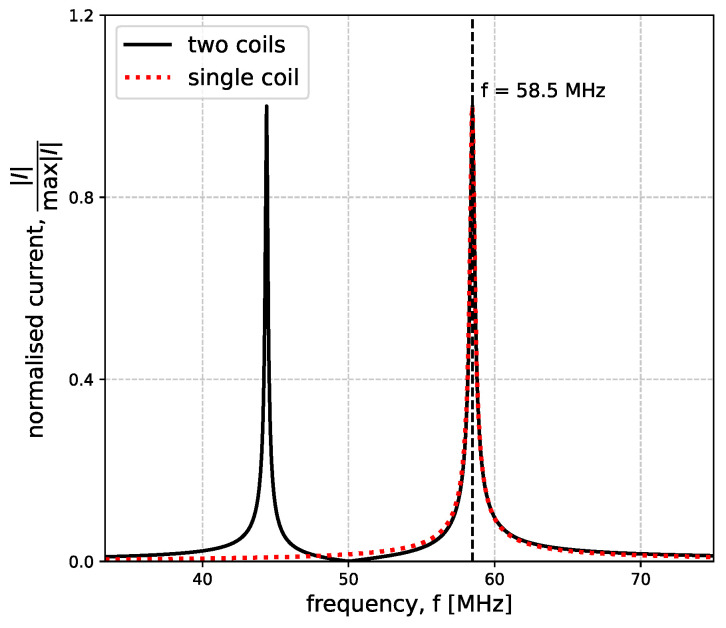
Absolute normalized current for a coil placed h=5 mm above a 10 cm thick conductor of $\sigma =10^{8}$ S/m (red dotted line) and for a system of two axially coupled coils placed 2h=10 mm apart (black line). The coils have $r_{0}=25$ mm, w=1 mm, $f_{0}=50$ MHz and $Q_{0}=250$.

**Figure 7 sensors-23-09711-f007:**
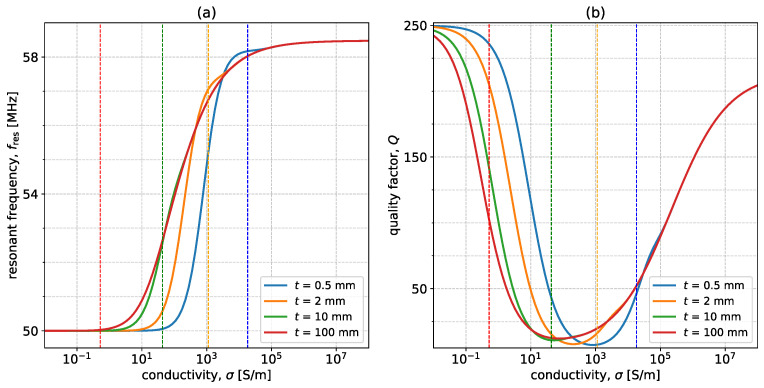
Variation of (**a**) $f_{\mathrm{res}}$ and (**b**) *Q* with conductivity, σ, for four values of thickness: 0.5 mm (blue), 2 mm (orange), 10 mm (green), and 100 mm (red). The parameters of the system are $r_{0}=25$ mm, w=1 mm, $f_{0}=50$ MHz, h=5 mm, $Q_{0}=250$, and $\mu_{r}=\varepsilon_{r}=1$. The dotted lines indicate positions where the skin depth is equal to the sample thickness.

**Figure 8 sensors-23-09711-f008:**
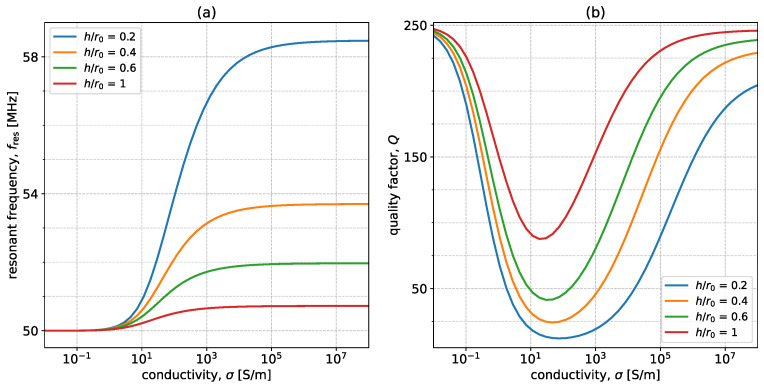
Variation of (**a**) $f_{\mathrm{res}}$ and (**b**) *Q* with σ for four values of $h/r_{0}$: 0.2 (blue), 0.4 (orange), 0.6 (green), and 1 (red). The other system parameters are w=1 mm, $f_{0}=50$ MHz, $Q_{0}=250$, t=10 cm, and $\mu_{r}=\varepsilon_{r}=1$.

**Figure 9 sensors-23-09711-f009:**
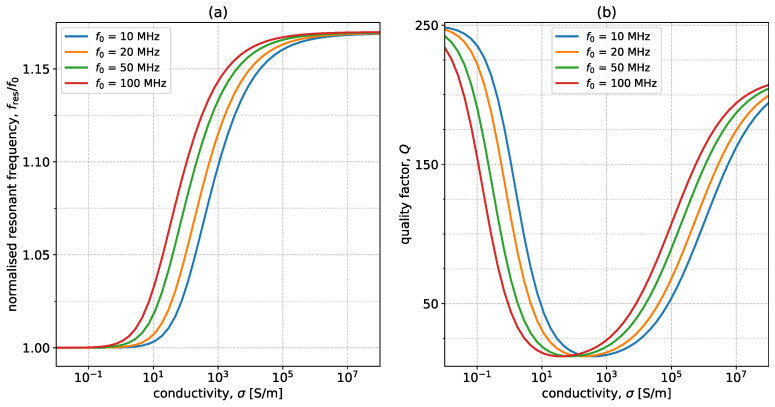
Variation of (**a**) normalized resonant frequency, $f_{\mathrm{res}}/f_{0}$, and (**b**) *Q* with σ for four values of free-space resonant frequency, $f_{0}$: 10 MHz (blue), 20 MHz (orange), 50 MHz (green), and 100 MHz (red). The other system parameters are $r_{0}=25$ mm, w=1 mm, h=5 mm, $Q_{0}=250$, t=10 mm, and $\mu_{r}=\varepsilon_{r}=1$.

**Figure 10 sensors-23-09711-f010:**
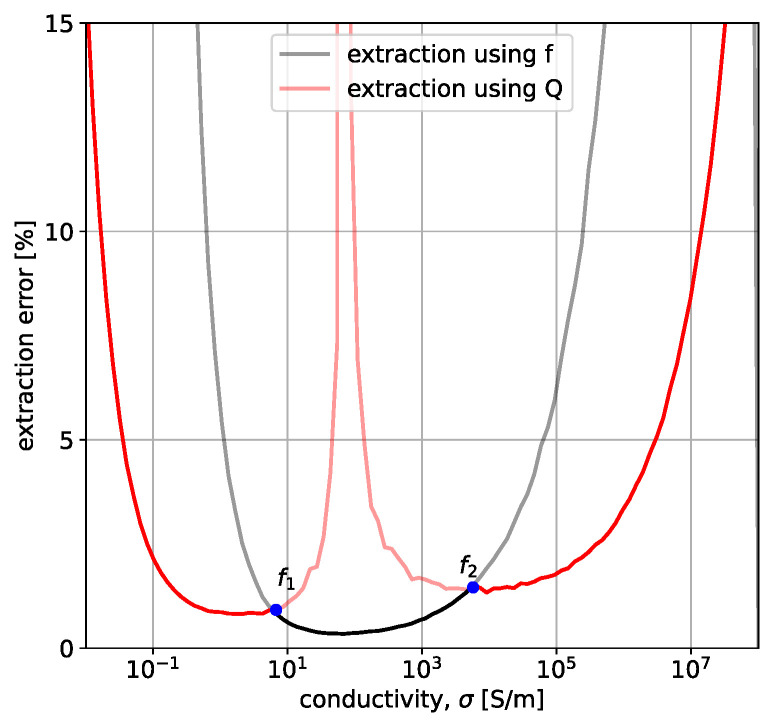
Extraction in the two presented regimes: $f_{\mathrm{res}}$ (black line) and *Q* (red line). The opaque lines represent areas where each method was chosen. $f_{1}$ and $f_{2}$ are cut-off frequencies for changing the selection criteria. The system parameters are $r_{0}=25$ mm, w=1 mm, $f_{0}=50$ MHz, h=5 mm, $Q_{0}=250$, t=10 cm, and $\mu_{r}=\varepsilon_{r}=1$. The artificial errors are 0.01% in $f_{\mathrm{res}}$ and 0.5% in *Q*.

**Figure 11 sensors-23-09711-f011:**
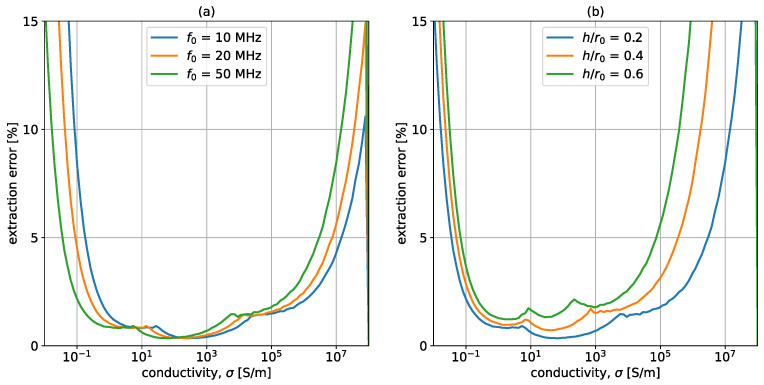
Errors in extraction for different (**a**) free-space frequencies, $f_{0}$, (**b**) height and coil radius ratios $h/r_{0}$. The artificial errors are 0.01% in $f_{\mathrm{res}}$ and 0.5% in *Q*. Unless otherwise specified in the legend, the system parameters are $r_{0}=25$ mm, w=1 mm, $f_{0}=50$ MHz, h=5 mm, $Q_{0}=250$, t=10 cm, and $\mu_{r}=\varepsilon_{r}=1$.

**Figure 12 sensors-23-09711-f012:**
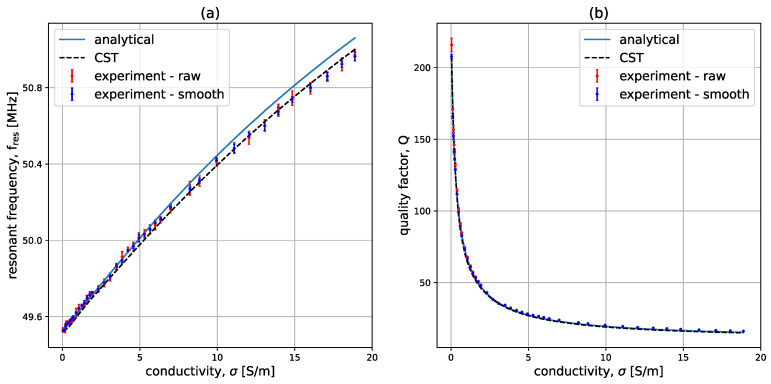
Variation of (**a**) $f_{\mathrm{res}}$, and (**b**) *Q* with σ in the analytical model (blue solid line), the raw experimental data (red data points), the smoothed experimental data (blue data points), CST Microwave Studio data (black dashed line). The system parameters are $r_{0}=25$ mm, w=1.04 mm, h=5.32±0.11 mm, $f_{0}=49.777$ MHz, $Q_{0}=256$, t=10 cm, $\mu_{r}=1$, and $\varepsilon_{r}=80$.

**Figure 13 sensors-23-09711-f013:**
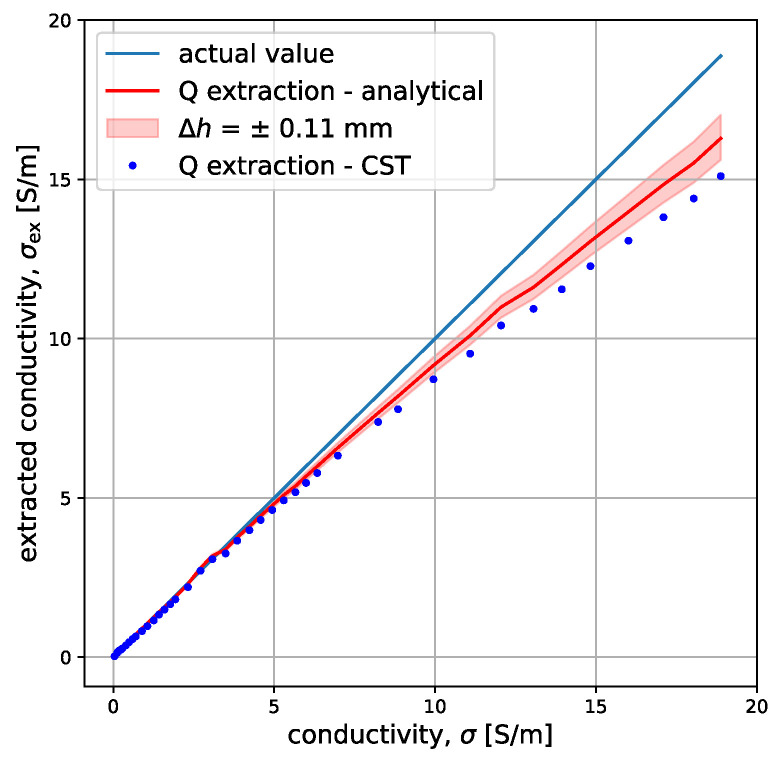
Extracted conductivity from the analytical model (red line) with the error bar for height uncertainty (red transparent area) and from CST data (blue dots). The parameters of the system are $r_{0}=25$ mm, w=1.04 mm, h=5.32±0.11 mm, $f_{0}=49.777$ MHz, $Q_{0}=256$, t=10 cm, $\mu_{r}=1$, and $\varepsilon_{r}=80$.

**Table 1 sensors-23-09711-t001:** Conductivity, σ, and relative permittivity, $\varepsilon_{r}$, for various body tissues at 50 MHz [49,50,51].

Tissue	σ [S/m]	$\varepsilon_{r}$
Blood	1.19	94.2
Muscle	0.68	77.1
Heart	0.65	118.0
Deflated Lung	0.52	81.3
Inflated Lung	0.28	41.3
Dry Skin	0.41	107.2
Fat	0.03	6.9

**Table 2 sensors-23-09711-t002:** Comparison between the mean error in conductivity extraction using the analytical model and the CST numerical results for different conductivity ranges.

Range	Nb. of Samples	Analytical Extraction	CST Extraction
σ < 1 S/m	**11**	**4.5%**	**9.2%**
σ < 5 S/m	**25**	**3.0%**	**7.1%**
σ < 12.5 S/m	**35**	**3.9%**	**7.9%**
σ < 20 S/m	**42**	**5.5%**	**9.7%**

## Data Availability

Data are contained within the article.

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
