# Peer review of "Design of a Remote, Multi-Range Conductivity Sensor"

_sensors, 2023, doi:10.3390/s23249711_

Round 1

Reviewer 1 Report

Comments and Suggestions for Authors

1. What effect does placing a coil near a material with different electromagnetic properties have on magnetic flux?

2. What is the recommended resonant frequency range for high, medium, and low conductivity samples?

3. What level of conductivity can be detected using this method, and what is its relevance to body imaging?

4. How do eddy currents affect the magnetic flux in the presence of a testing material?

5. How is the quality factor used as a separation criterion, and what role does it play in separating low σ samples?

6. How does the performance of the analytical model compare to CST data in terms of conductivity extraction?

7. What is the effect of lower free-space resonant frequencies on the detection of large conductivities, and what is the error threshold mentioned?

Reviewer 2 Report

Comments and Suggestions for Authors

Dear Authors,

thank you for this interesting study. I appreciate and value all your findings, although I must admit that especially on page 9 and 10 a sketch and some naming of geometry parameters would have had helped significantly in the understanding. I strongly recommend to introduce all geometrical parameters (e.g. "slab thickness") in some technical sketch representing your real-life setup. On top a simple photography would be super helpful to get a better understanding of your technical setup and arrangement you used for your measurements. Please do not get me wrong, the research is very fascinating and I like your findings, the more frustrating it is if you have to carefully decode the technical setup and do not feel the trust in the experimental arrangement as basic information are hidden in the text or not presented. A drawing, a photography, a higher systematic in the text when referring to geometries instead of using unprecise naming of geometrical shapes not introduced sorrowly, and this paper jumps from good to excellent! And - at least on my copy - figure 2 only shows the caption, not content.

Comments on the Quality of English Language

no improvements required

Reviewer 3 Report

Comments and Suggestions for Authors

1- The text of the manuscript should be reviewed for spelling, grammar, and spelling.

2- The number of related references is incomplete. It is recommended to add several new articles with details to the manuscript.

You can use the following articles:

DOI: 10.1016/j.apmt.2023.101877

DOI: 10.4218/etrij.2018-0013

DOI: 10.1109/TRS.2023.3320869

3- The most important concern is that this structure should be built. The measurement results are presented and compared. Add sensitivity and error.

4- It is recommended to check and report the characteristic of SAR in different situations.

5- Please report the permeability, permittivity, and reflex coefficient.

Comments on the Quality of English Language

 The text of the manuscript should be reviewed for spelling, grammar, and spelling.

Round 2

Reviewer 1 Report

Comments and Suggestions for Authors

The author has fully responded to the question.

Reviewer 3 Report

Comments and Suggestions for Authors

This version of the manuscript has better conditions. Therefore, it can be considered as an option for publication.